# Intelligent maintenance model for battery energy storage system

**Yufei Zhuang (2024319242)**
Department of Computer Science
zhuangyf24@mails.tsinghua.edu.cn

## Abstract

Battery thermal runaway has short evolution time and strong destructiveness. This project focuses on early battery fault diagnosis and early warning. This project mainly constructs a large battery fault early warning model based on methods such as statistical analysis, machine learning, data-driven models, and expert knowledge rules to help solve the safety problem of battery energy storage systems.

## 1  Background

Electrochemical energy storage technology realizes the storage, release or rapid power exchange of electric energy through various batteries. It can smooth the fluctuation of new energy power generation, improve the balance of power supply and demand, and enhance the stability of power systems. It is mainly applied in fields such as power, energy, and transportation and is one of the key technologies supporting a new power system with new energy as the main body.

Lithium-ion battery technology has developed rapidly in recent years due to its characteristics of high energy density, long cycle life, and high conversion efficiency. At present, the installed capacity of lithium-ion battery energy storage in China accounts for more than 80% of the overall scale of electrochemical energy storage. Since lithium-ion batteries involve complex electrochemical reactions, their performance will rapidly decline under certain working conditions or when improper operation is used, and even thermal runaway may occur, leading to safety accidents, which pose a serious hidden danger to people's lives and property safety.

Problems such as imperfect daily operation and maintenance systems and defects in battery protection systems are important reasons for accidents. Timely mastering the operating status of equipment, formulating targeted operation and maintenance strategies, and giving early warnings in time when early fault characteristics appear in batteries are particularly important for the safe and stable operation of lithium-ion battery energy storage systems.

## 2  Definition

### 2.1  SOC

Battery SOC refers to the available state of the remaining charge in the battery, generally expressed as a percentage. It is the ratio of the remaining charge margin of the battery to the nominal (rated) charge capacity of the battery.

### 2.2  SOH

SOH represents the ability of the current battery to store electrical energy relative to a new battery and is an index for evaluating the aging degree of the battery's health status.

# 3   Related Work

At present, the research on the operation and maintenance of lithium-ion battery energy storage systems at home and abroad is mainly divided into two categories:

First, online evaluation of battery status. This method can timely detect the early decline of battery performance through online monitoring of battery state of charge (SOC), state of health (SOH), etc., and has a certain role in preventing battery cell or module failures.

Second, fault diagnosis and early warning of energy storage systems. This method designs a fault diagnosis system by using big data and artificial intelligence technologies such as fault trees, expert systems, and machine learning to quickly identify the cause of faults after the energy storage system fails or realize early warning of energy storage system faults.

# 4   Proposed Method

## 4.1   Research Goal

In this project, research will mainly be conducted in the direction of battery fault diagnosis and early warning. Once a battery experiences thermal runaway, it has characteristics such as short evolution time, strong destructiveness, and difficulty in blocking. Research on battery fault diagnosis mainly focuses on early fault diagnosis of batteries. For example, by measuring battery voltage and temperature online to estimate the internal state of the battery (such as equivalent short-circuit resistance), early spontaneous internal short-circuit diagnosis of the battery can be achieved.

When a fault occurs in a lithium-ion battery energy storage system, the diagnosis system can realize system fault diagnosis based on the difference in external characteristics between when there is a fault and when it is normal. The main processes of fault diagnosis include fault feature extraction, fault isolation and estimation, and fault evaluation and decision-making.

## 4.2   Fault diagnosis method based on a battery model

For the fault diagnosis method based on a battery model, the key is to establish an accurate and reliable battery model. By comparing the difference between the model predicted value and the measured value, fault early warning can be achieved. The equivalent circuit model is a semi-empirical and semi-mechanistic model with low computational complexity and excellent prediction performance. It has been widely used in battery management systems. The diagnosis method integrating parameter estimation estimates the model parameters through online data and then diagnoses battery faults according to the model parameters. The diagnosis method integrating state estimation calculates the estimated values of battery terminal voltage and temperature through filtering algorithms and diagnoses battery faults according to the residuals between the voltage and temperature estimated values and the measured values. A battery fault model can also be established. By comparing whether the measured value matches the predicted value of the fault model, the diagnosis of such faults can be achieved.

## 4.3   Fault diagnosis method without a battery model

The diagnosis method without a battery model does not need to model the dynamic characteristics of a single battery. Online implementation can avoid parameter update and iteration of a single battery model and has high efficiency. However, compared with the model-based fault diagnosis method, this kind of method depends on the quality of sample data and has relatively poor extrapolation ability, and it is often difficult to give quantitative diagnosis results. The diagnosis method based on statistical analysis directly uses the current, voltage, and temperature data obtained by the signal acquisition system and analyzes them using statistical methods such as information entropy and normal distribution. By setting abnormal coefficients or thresholds, battery fault diagnosis can be achieved. The diagnosis method based on data-driven and large models realizes fault diagnosis and life prediction of batteries such as over-temperature, low-temperature, overcharging, and over-discharging by establishing a diagnosis system that connects the external electrical characteristics and internal chemical mechanisms of batteries. Commonly used methods include neural networks, support vector regression, fuzzy logic, Bayesian theory, etc. The expert knowledge rules based on

battery system fault feature analysis realize battery fault diagnosis through historical operation data and fault maintenance record information.

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
